

# Circulating cell-free DNA-based methylation pattern in plasma for early diagnosis of esophagus cancer

Rui Wang[1], Yue Yang[2], Tianyu Lu[2], Youbin Cui[2], Bo Li[1] and Xin Liu[2]

[1] School of Public Health, Jilin University, Changchun, Jilin, China
[2] Department of Thoracic Surgery, First Hospital of Jilin University, Changchun, Jilin, China

## ABSTRACT

With the increased awareness of early tumor detection, the importance of detecting and diagnosing esophageal cancer in its early stages has been underscored. Studies have consistently demonstrated the crucial role of methylation levels in circulating cell-free DNA (cfDNA) in identifying and diagnosing early-stage cancer. cfDNA methylation pertains to the methylation state within the genomic scope of cfDNA and is strongly associated with cancer development and progression. Several research teams have delved into the potential application of cfDNA methylation in identifying early-stage esophageal cancer and have achieved promising outcomes. Recent research supports the high sensitivity and specificity of cfDNA methylation in early esophageal cancer diagnosis, providing a more accurate and efficient approach for early detection and improved clinical management. Accordingly, this review aims to present an overview of methylation-based cfDNA research with a focus on the latest developments in the early detection of esophageal cancer. Additionally, this review summarizes advanced analytical technologies for cfDNA methylation that have significantly benefited from recent advancements in separation and detection techniques, such as methylated DNA immunoprecipitation sequencing (MeDIP-seq). Recent findings suggest that biomarkers based on cfDNA methylation may soon find successful applications in the early detection of esophageal cancer. However, large-scale prospective clinical trials are required to identify the potential of these biomarkers.

## INTRODUCTION

Esophageal cancer (EC) is a globally prevalent malignancy. According to the GLOBOCAN (2020) data (*Sung et al., 2021*), EC ranks eighth in terms of incidence and sixth in terms of cancer-related mortality. Predictions based on GLOBOCAN (2020) data by *Morgan et al. (2022)* suggest that there will be 957,000 new cases and 880,000 deaths from EC in 2,040. Developing countries account for approximately 80% of new EC cases, with 60% occurring in China (*Ferlay et al., 2010*). China is a high-risk country for EC (*Zhao et al., 2012*; *Chen et al., 2016*), with an estimated 480,000 new cases and 370,000 deaths annually (*Zhou et al., 2019*). Age and regional differentiation are the most significant characteristics of EC in China, with a higher incidence among males and in rural areas (*Liang, Fan &*

Corresponding authors
Bo Li, li_bo@jlu.edu.cn
Xin Liu, xin_liu21@mails.jlu.edu.cn

*Qiao, 2017*; *Chen et al., 2023*). Due to delayed diagnosis, surgical resection often cannot provide a cure. In China, the survival rate of patients with late-stage esophageal squamous cell carcinoma (ESCC) is less than 10%, while early detection leads to an 85% survival rate (*Wang et al., 2004*). However, cancer screening still relies on non-molecular techniques with low specificity and sensitivity. Currently, esophageal mucosal iodine (Lugol's) staining endoscopy is the gold standard for diagnosing abnormal esophageal mucosal lesions (*Liang, Fan & Qiao, 2017*; *Piñerúa Gonsálvez, Zambrano-Infantino & Benítez, 2019*). Liquid biopsy for tumors, with its simple procedure, high sensitivity and specificity, and noninvasive or minimally invasive features, has revolutionized traditional cancer treatment by dynamically monitoring the onset, development, and metastasis of the disease by detecting relevant tumor markers in patients' blood (*Peneder et al., 2021*).

Numerous studies have demonstrated the potential of circulating cell-free DNA (cfDNA) as a biomarker for early cancer diagnosis, making it a valuable liquid biopsy analyte (*Freitas et al., 2021*; *Jiang et al., 2022*). For instance, *Egyud et al. (2019)* showed the dynamic potential of cfDNA as a biomarker for monitoring treatment response and disease recurrence in patients with esophageal adenocarcinoma (EDAC). Likewise, *Azad et al. (2020)* found that cfDNA in blood samples from patients undergoing chemoradiotherapy for EC was associated with tumor progression, metastasis, and disease-specific survival. The methylation features of cfDNA are critical epigenetic modifications that exhibit significant differences between healthy individuals and those with various diseases, particularly malignant tumors (*Klein et al., 2021*). Moreover, prospective studies, such as Circulating Cell-Free Genome Atlas (CCGA) conducted by GRAIL (NCT02889978 and NCT3085888) confirmed the high specificity and sensitivity of cfDNA methylation for the early detection of various cancers (*Liu et al., 2020a*).

Numerous scholars have reviewed the applications of cfDNA in the early diagnosis of various tumors, such as central nervous system tumors and head and neck cancers (*McEwe, Leary & Lockwood, 2020*; *Birknerova et al., 2022*). Furthermore, some researchers have meticulously summarized the applications of cfDNA in directing adjuvant therapy for EC (*Salati et al., 2021*). However, there remains a gap in this field, as only a few scholars have explored the potential applications of cfDNA methylation in the early diagnosis of esophageal cancer. This article aims to review the latest advancement in cfDNA detection and its application in the early detection of EC. First, the biological characteristics of cfDNA are outlined. Then, the latest research on cfDNA methylation biomarkers for early detection of EC is summarized, along with the currently available sequencing methods for cfDNA methylation. Finally, the clinical utility, limitations, and future development directions of cfDNA in the early detection of EC are discussed.

## SURVEY METHODOLOGY

The PubMed database was utilized to conduct a literature search related to the keywords "cell-free DNA", "circulating tumor DNA", "DNA methylation", "esophageal cancer", "cancer", and "early diagnosis". Subsequently, we collated the retrieved articles, including those cited in the recovered articles. Approximately 1,100 related articles were thoroughly read between January 2018 and May 2022.

### The rationale for why it is needed

Numerous studies have extensively reviewed the application of cfDNA in the early diagnosis of various tumors, including liver and lung cancers. Furthermore, some researchers have provided comprehensive summaries of the use of cfDNA as adjuvant therapy for esophageal cancer. However, insufficient evidence exists as only a few scholars have reviewed the potential applications of cfDNA methylation in the early diagnosis of EC.

This review aims to provide a comprehensive overview of methylation-based cfDNA research and the latest advancements in the early detection of EC. Thus, researchers can gain a preliminary understanding of the relevance of cfDNA methylation and its application in the early detection of EC. Finally, we address the clinical applications, limitations, and future directions of cfDNA in the early detection of EC. This enables readers to comprehend the limitations and potential areas for further development in this research field, thereby promoting significant advancements in scientific achievements within this domain.

### The audience it is intended for

Doctors specializing in oncology, thoracic surgery, and clinical laboratory medicine may find this study intriguing. The integration of cfDNA methylation with machine learning has accelerated the transition of this field into clinical applications. Ultimately, as cfDNA detection technology continues to advance and becomes implemented in clinical practice, patients can anticipate reduced treatment costs and significantly improved survival rates.

### cfDNA and ctDNA biology

cfDNA refers to fragments of chromosomal material released due to cell death and is present in the circulatory system (*Lo et al., 2010*; *Heitzer, Auinger & Speicher, 2020*). Typically, these double-stranded fragments are approximately 150–200 base pairs in length (*Warton et al., 2014*). In healthy individuals, cfDNA mainly originates from hematopoietic cells; however, its composition may change under certain physiological or pathological conditions. This difference has been exploited for noninvasive liquid biopsy, with fetal-specific cfDNA used for prenatal diagnosis (*Lo et al., 1997*; *Chiu et al., 2011*; *Hou et al., 2012*) and tumor-specific cfDNA used for cancer diagnosis at different stages (*Abbosh et al., 2017*; *Dasari et al., 2020*; *Nakamura et al., 2021*; *Herberts et al., 2022*). Under normal physiological conditions, cfDNA concentrations in healthy individuals typically range from 1–50 ng/mL, whereas cfDNA concentrations in patients with tumors can exceed 1,000 ng/mL (*Meddeb et al., 2019*; *Osumi et al., 2019*). The genetic information contained in cfDNA can accurately detect tumor mutations and gene expressions (*Esfahani et al., 2022*), and its short half-life (16 min to 2 h) enables dynamic tracking of cancer progression (*Lo et al., 1999*; *Yu et al., 2013*).

ctDNA, released by tumor cells, is a type of cfDNA biomarker that carries tumor-specific genetic and epigenetic abnormalities. It serves as a potential substitute for tumor tissue DNA for diagnosing and monitoring prognostic changes (*Jin et al., 2020*; *Yang et al., 2020*; *Zhou et al., 2021*; *Li et al., 2022*). However, despite the promise of ctDNA mutation detection, its sensitivity and accuracy in detecting early-stage cancer remain low (*Cohen et al., 2018*). Abnormal changes in DNA methylation, including high CpG

island (CGI) methylation (*Weinberg et al., 2021*; *Ren et al., 2022*), are among the earliest molecular changes in cancer progression. *Loyfer et al. (2023)* recently published the most comprehensive human single-cell-type DNA methylation atlas to date, showing remarkable consistency in the DNA methylation patterns of the same cell type among different healthy individuals. Based on the highly conserved DNA methylation patterns of the same cell type among individuals, *Loyfer et al. (2023)* studied cell-specific differentially methylated regions to detect the content of specific cell types within cfDNA mixtures. Several studies have examined early cancer detection based on ctDNA methylation features (*Luo et al., 2020*; *Papanicolau-Sengos & Aldape, 2022*).

The final concentration of ctDNA can be influenced by several factors, such as tumor volume, location, and vascular formation, as well as anti-tumor treatments, such as surgery, chemotherapy, and radiotherapy. Liver and kidney clearance rates also affect ctDNA concentration (*Bettegowda et al., 2014*). Moreover, ctDNA concentration can be affected by other conditions such as trauma, myocardial infarction, stroke, and chronic diseases such as diabetes and inflammation. Optimal fluid selection also plays a crucial role in improving ctDNA detection. For instance, monitoring ctDNA in cerebrospinal fluid can reflect the disease status of patients with brain metastases (*Wu et al., 2023*), and using urine can facilitate noninvasive liver cancer screening (*Kim et al., 2022*).

Given the complexity of cfDNA and ctDNA, several factors must be considered when conducting liquid biopsy-related research. In addition, various techniques and detection instruments, along with pre-analytical factors, are essential for the comprehensive and accurate detection and analysis of all circulating DNA.

## Pre-analytical and analytical phase examination

The pre-analytical workflow begins with the selection of an appropriate sample type, with plasma being a clinically convenient and compliant option. Currently, blood-based detection is the standard for multiple cancers (*Abbosh, Birkbak & Swanton, 2018*; *Luo et al., 2020*; *Ignatiadis, Sledge & Jeffrey, 2021*). However, there are several factors to consider during the pre-analysis of ctDNA, such as blood vessel selection, processing delay, centrifugation protocols, sample transportation, storage conditions, and anticoagulant selection, all of which affect the concentration of ctDNA (*Jen, Wu & Sidransky, 2000*; *Sozzi et al., 2005*; *Leest et al., 2020*; *Lehle et al., 2023*). In 2022, European Society for Medical Oncology (EMSO) (*Pascual et al., 2022*) released recommendations for ctDNA detection technology that include the careful selection of blood collection times based on clinical conditions, the choice of blood collection tubes based on ctDNA processing time and detection method, and the long-term storage of plasma samples at $-80\ °C$ to reduce repeated freeze-thaw cycles and minimize temperature changes.

Even with centrifugation and purification, there are still challenges in the process, such as the loss of a significant amount of DNA samples during purification and the potential contamination of tumor samples with DNA from blood cells during centrifugation (*Leest et al., 2020*; *Lehle et al., 2023*). To address these challenges, using suitable commercial kits, such as those from QIAamp, Microdiag®, and the MicroDiag® EGFR gene mutation detection kit, can enhance DNA purification and concentration (*Wang et al., 2021*).

Additionally, the Oxford Nanopore (ONT) platform provides a portable method for rapid genomic sequencing that can analyze DNA methylation without the need for complex sample processing, presenting new opportunities for real-time sequencing (*Katsman et al., 2022*).

## Methods in cfDNA methylation analysis

The main methods of DNA methylation sequencing include whole-genome bisulfite sequencing (WGBS), reduced-representation bisulfite sequencing (RRBS), and methylated DNA immunoprecipitation sequencing (MeDIP-seq). WGBS is a technique commonly used to determine DNA methylation levels in the genome (*Jammula et al., 2020*; *Rajamäki et al., 2021*; *Tanaka et al., 2021*). It relies on two sequencing strategies: library construction and sequencing data, with various sequencing technologies and data analysis used to analyze the sequencing data. The most commonly used whole-genome sequencing technologies for WGBS include Illumina, PacBio, and Oxford Nanopore Technologies (*Gouil & Keniry, 2019*; *Alfaro et al., 2021*; *Zee et al., 2022*). RRBS is a newer-generation sequencing method with several advantages over WGBS. It uses approximately 5% of the genomic loci for sequencing analysis (*Liu et al., 2020b*; *Gu et al., 2021*; *Sharma et al., 2022*), resulting in reduced sequencing depth and coverage requirements, lower sequencing costs, and shorter sequencing times. RRBS also has high selectivity and accuracy, enabling the detection of DNA methylation status at single-base resolution in CpG sites. MeDIP-seq is a high-throughput sequencing-based epigenomic technology that immunoprecipitates methylated DNA fragments using specific antibodies before sequencing analysis (*Shen et al., 2019*; *Zhang et al., 2022*). This method provides high sequencing depth and coverage, enabling comprehensive analysis of DNA methylation across the genome, including CpG islands and non-CpG island regions. MeDIP-seq is particularly useful for identifying DNA methylation biomarkers and exploring the relationship between DNA methylation and gene regulation (*Zhou et al., 2018*; *Beck, Ben Maamar & Skinner, 2022*). However, these methods generate a large amount of information, leading to numerous differentially methylated regions (DMRs). However, they may have insufficient coverage and sequencing depth in certain regions, resulting in missing or false-positive results (*Ben Maamar et al., 2021*; *Gong et al., 2022*). Therefore, further validation of the selected DMRs is necessary. Targeted region methylation resequencing (Hi-MethylSeq), also known as bisulfite amplicon sequencing (BSAS), can accurately quantify methylation levels in multiple regions and loci of candidate genes in large populations based on WGBS, RRBS, and other studies. Thus, the Hi-MethylSeq technology serves as a powerful tool for subsequent validation of WGBS (*Cai et al., 2021*).

With the rapid development of sequencing technologies, there has been a continual improvement in sequencing quality. However, certain limitations still persist, such as the inability to directly identify the methylation states and sites of genes, which presents many challenges when analyzing gene expression, DNA modifications, and related aspects. To address these limitations, researchers have developed a methylation-sensitive restriction enzyme-based sequencing (MRE-seq) technology focused on gene enrichment. This technology enhances the size and concentration of DNA fragments by combining

methylation-sensitive restriction enzymes with high-throughput sequencing technology. Ultimately, it enables the comparison of the methylation states between different samples. For example, MRE-seq technology enables the sequencing of methylation sites in the DNA of both cancer and normal cells, allowing for a comparison of methylation status in different states and contributing to the study of the mechanism underlying cancer development (*Shin et al., 2022*). However, MRE-seq technology is limited to sequences containing CG sites and cannot directly detect other methylation sites, such as m6A. Therefore, it is necessary to employ complementary detection technologies to investigate the methylation status at these sites (*Sun, Wu & Ming, 2019*). Compared to sequencing methods that require specific enzyme cleavage and chemical reactions to determine DNA methylation status, the newly developed single-molecule real-time sequencing (SMRT) technology by the US-based PacBio company can directly achieve high-precision, high-throughput detection of DNA methylation status. This is achieved by monitoring the single-molecule long-chain amplification process of DNA polymerase on DNA templates while retaining the original DNA methylation status (*Forde et al., 2019*; *Chen et al., 2022*). In addition, nanopore sequencing technology reads DNA sequence information by measuring the transient electrical current changes in single DNA molecules passing through a nanopore (*Liu et al., 2021b*). This technology has a higher resolution and accuracy, can analyze longer DNA fragments for tasks such as genome-wide methylation spectrum analysis, and has a higher sensitivity for detecting low-frequency methylation sites (*Tourancheau et al., 2021*; *Katsman et al., 2022*). Recently, scholars such as *Yu et al. (2023)* compared the advantages and disadvantages of SMRT technology and nanopore sequencing technology across different long-read sequencing platforms. SMRT technology, based on a specialized chip for SMRT cells, limits the number of single reaction layers by restricting the position of the fixed DNA polymerase reaction. This restriction, in turn, limits the throughput and read length but generates data with a higher percentage of long cfDNA. In contrast, nanopores use a single-molecule electrical transfer chip with a nanopore, making the throughput and read length unlimited. Finally, we summarize recent clinical studies on the application of common sequencing techniques for cfDNA methylation in cancer research (*Bruzek et al., 2020*; *Zhang et al., 2020*; *Berchuck et al., 2022*; *Choy et al., 2022*; *Marinelli et al., 2022*) (Table 1).

## Clinical application of cfDNA methylation in early detection of EC

Given the practicality of early cancer screening, analyzing cfDNA using simple and highly specific blood sampling may be more advantageous than traditional screening tools. It enables the analysis of tumors that are undetectable or uncertain using imaging techniques. Current research on cfDNA as a cancer biomarker has primarily focused on mutation detection, atypical fragment patterns, and abnormal methylation (*Zviran et al., 2020*; *Liu et al., 2021a*; *Esfahani et al., 2022*). In a CCGA study, *Jamshidi et al. (2022)* compared ten machine learning classifiers using various cfDNA features and found that cfDNA methylation patterns were the most promising for the early detection of various cancers. In this review, we focused on the research progress regarding cfDNA methylation-based biomarkers for early EC detection (Fig. 1).

**Table 1** Clinical applications of common methylation sequencing methods in tumors.

| Author | Title | Sequencing method | Subject | Findings |
|---|---|---|---|---|
| *Zhang et al. (2020)* | Hypomethylation in HBV integration regions aids non-invasive surveillance to hepatocellular carcinoma by low-pass genome-wide bisulfite sequencing | WGBS | The methylation of hepatitis B virus integration regions and genome distribution of cfDNA | 1. Methylation levels of integration sites certain candidate regions exhibited an area under the AUC value >0.85 to discriminate HCC from non-HCC samples; 2. The validation cohort achieved the prediction performance with an AUC of 0.954. |
| *Marinelli et al. (2022)* | Methylated DNA markers for plasma detection of ovarian cancer: discovery, validation, and clinical feasibility | RRBS | 11-MDM panel highly discriminated OC from controls | 96% (95% CI (89–99%)) specificity; 79% (69–87%) sensitivity, and the AUC 0.91 (0.86–0.96). |
| *Berchuck et al. (2022)* | Detecting neuroendocrine prostate cancer through tissue-informed cell-free DNA methylation analysis | MeDIP-seq | NEPC Risk Score | Applying the predefined NEPC Risk Score cutoff to the validation cohort resulted in 100% sensitivity and 95% specificity for detecting NEPC. |
| *Choy et al. (2022)* | Single-molecule sequencing enables long cell-free DNA detection and direct methylation analysis for cancer patients | Single-molecule real-time sequencing | HCC methylation score | The use of long cfDNA molecules demonstrated greatly discriminatory power (AUC: 0.91) |
| *Bruzek et al. (2020)* | Electronic DNA analysis of CSF cell-free tumor DNA to quantify multi-gene molecular response in pediatric high-grade glioma | Oxford Nanopore Technology | CSF cf-tDNA variant allele fraction | Nanopore demonstrated 85% sensitivity and 100% specificity in CSF samples with 0.1 femtomole DNA limit of detection and 12-hour results. |

Notes.
HBV, hepatitis B virus; HCC, hepatocellular carcinoma; cfDNA, cell-free DNA; AUC, receiver operation curve; MDM, methylated DNA marker; OC, ovarian cancer; NEPV, neuroendocrine prostate cancer; CSF, cerebrospinal fluid; cf-tDNA, cell-free tumor DNA; WGBS, whole genome bisulfite sequencing; RRBS, reduced representation bisulfite sequencing; MeDIP-seq, methylated DNA immunoprecipitation sequencing.

DNA methylation is an important epigenetic modification that affects gene expression, genomic stability, and development. It has been widely used to assess cancer occurrence, progression, and treatment response (*Wu & Zhang, 2014*). Cytosine methylation (5-methylcytosine, 5mC) is a well-recognized epigenetic modification that affects gene expression. Reconstruction of DNA 5mC has been widely used to study cancer occurrence, progression, and treatment response (*Hu et al., 2021*). With the discovery of DNA demethylase-DNA dioxygenase (TET), the oxidative form of cytosine methylation modification has gained considerable attention (*Ito et al., 2011*). 5-hydroxymethylcytosine (5hmC) is the most common oxidative form of methylcytosine. It serves as an intermediate actively involved in demethylation and a stable modification form in the genome (*Bachman et al., 2014*; *Klungland & Robertson, 2017*). Many studies have confirmed that DNA modifications, such as 5mC and 5hmC, can be used as ideal biomarkers for cancer diagnosis (*Xiao et al., 2021*; *Zhang et al., 2021*; *Sjöström et al., 2022*; *Turpin & Salbert, 2022*). However, our study focused on the epigenetic modification of DNA methylation.

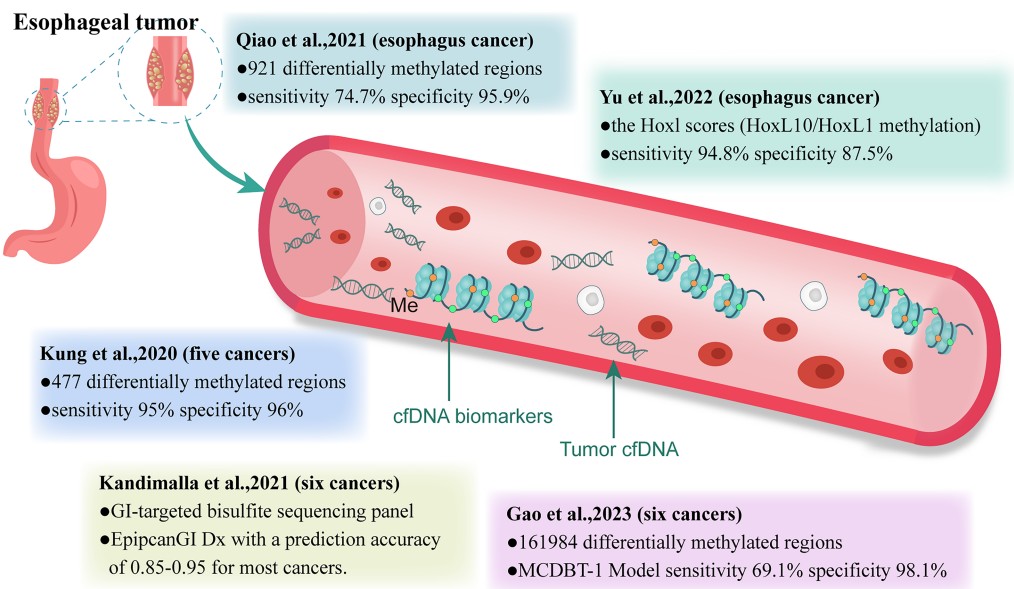

**Figure 1** **Clinical study of methylation-based cfDNA detection in cancer early screening.** GI, gastrointestinal; Me, methylation.

The advantages of DNA methylation analysis over gene mutation analysis in cancer detection are as follows (Fig. 2): First, it has higher clinical sensitivity and a broader dynamic range, as there are many methylation targets in tumors and multiple altered CpG sites in each targeted genomic region. However, only a fraction of mutations in cancer tissue can be detected in circulating cfDNA (*Garcia et al., 2019*). Second, compared to the highly individualized and heterogeneous nature of gene mutations, which do not provide accurate tumor origin or specific organ information (*Shahbandi, Nguyen & Jackson, 2020*; *Maxwell et al., 2022*), different tissue-derived cfDNA presents distinct methylation patterns, enabling tissue tracing (*Baylin & Jones, 2011*). Finally, the number of methylation sites is significantly larger than the number of point mutations. Analyzing a cluster of CpG sites known as a methylation block (MB) as a complete unit can result in stronger methylation signals through both lateral patterns and longitudinal abundance (*Guo et al., 2017*).

*Boldrin et al. (2020)* conducted a study that analyzed the methylation status of long interspersed nucleotide element (LINE-1) sequences in 21 circulating cfDNA, 19 esophageal adenocarcinoma (EADC), and two Barrett's esophagus samples. They also performed a longitudinal analysis of two patients with Barrett's esophagus and one patient with EADC. This study revealed low levels of methylation of LINE-1 sequences in EADC cfDNA. Additionally, the longitudinal analysis indicated a correlation between the methylation status of LINE-1 sequences in cfDNA and the progression to EADC (*Boldrin et al., 2020*).

Several other studies have investigated the role of DNA methylation in the development and early detection of EC (*Wang et al., 2020*). *Fan et al. (2022)* confirmed that the frequency of P16 methylation increases with the severity of esophageal lesions and that it can serve

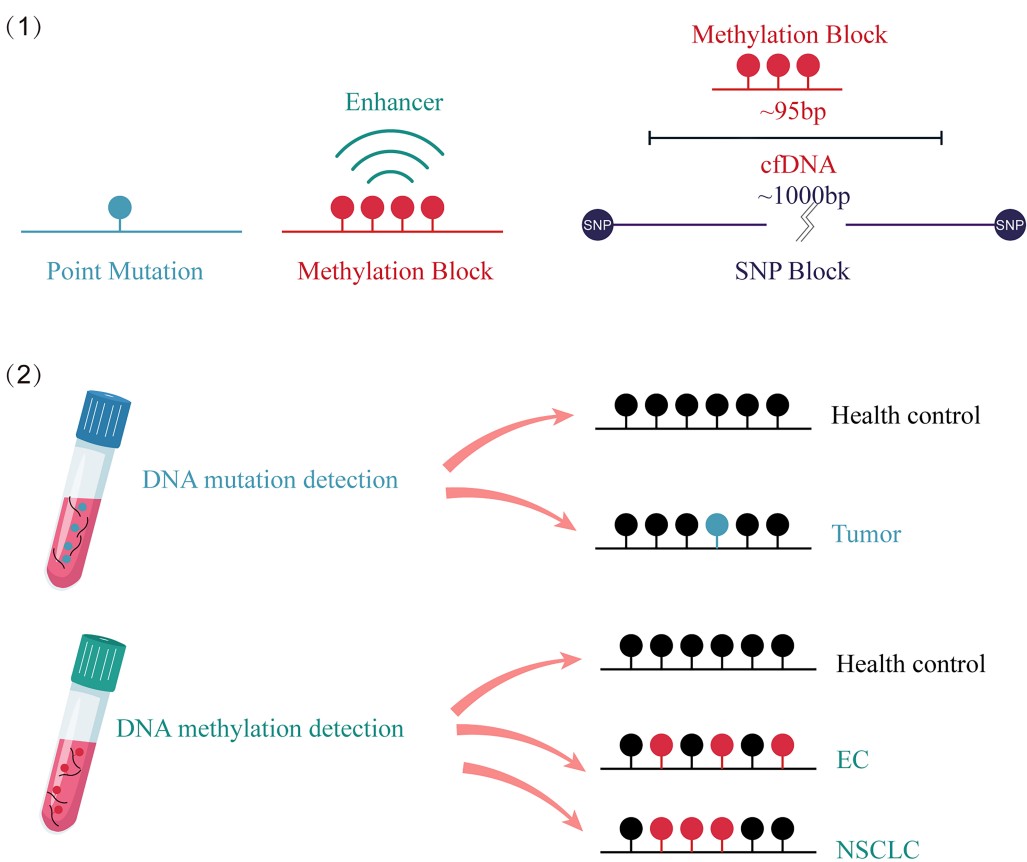

**Figure 2** **Comparison of the advantages between cfDNA methylation detection and cfDNA mutation detection.** (1) Compared to point mutations, clustered and amplified signals are observed in cfDNA methylated sites. (2) cfDNA mutation detection can distinguish between cancer patients and healthy individuals. cfDNA methylation detection not only identifies cancer patients, but also traces the origin of the cancer. EC, esophagus cancer; NSCLC, non-small-cell lung cancer; SNP, single nucleotide polymorphism; cfDNA, cell-free DNA.

as a biomarker for the early detection of ESCC and its precursor lesions. Scholars from Thailand have also found that P16 and TP53 methylation can serve as potential biomarkers for predicting EC, especially in individuals with drinking and smoking habits that can promote abnormal DNA methylation (*Poosari et al., 2022*). *Nasrollahzadeh et al. (2021)* compared the consistency of TP53 mutations in tumor tissues and cfDNA extracted from serum archives (the main source of cfDNA in retrospective studies) of 42 cases from high-risk areas in Iran and 39 matched controls. The results demonstrated a consistency of 24–36% between the variants detected in cfDNA from archived serum and paired FFPE ESCC tumor tissue, emphasizing the feasibility of early EC diagnosis through cfDNA detection (*Nasrollahzadeh et al., 2021*).

*Qiao et al. (2021)* developed an early screening model for EC based on cfDNA methylation. In their study, 161,984 cancer-related CpG sites were obtained from publicly available data in the TCGA and GEO databases, as well as internal data. Subsequently, 921 esophageal cancer-specific differentially methylated regions (DMRs) were identified and

optimized from esophageal tumors and their paired adjacent tissues and used to build an early screening model. This model was developed and tested on training and testing sets using plasma samples. Analyzing the model's performance on the training and testing sets showed a specificity of 95.2% for healthy individuals, 80% for benign esophageal diseases, and sensitivities ranging from 60%, 77.8%, 86.4%, to 100.0% for patients with EC at stages 0–IV, respectively (*Qiao et al., 2021*). These results preliminarily demonstrate the feasibility of liquid biopsy methylation detection for early screening of EC. To further validate the model's performance, *Qiao et al. (2021)* conducted a prospective, single-blind trial. In the independent validation group, the sensitivities for early (stages 0–II) and advanced (stages III–IV) esophageal cancers were 58.8% and 100.0%, respectively. Therefore, for patients highly suspected of having EC, it is worth considering whether a cfDNA methylation-based early screening model could replace commonly used auxiliary examinations as the primary diagnostic tool.

*Yu et al. (2022)* used whole-genome bisulfite sequencing (WGBS) to detect methylation features in ESCC and paired normal samples. Receiver operating characteristic curve analysis indicated that HOXC10 and HOXD1 methylation values were the best predictors for distinguishing ESCC samples from normal samples, achieving an AUC of 0.85. Consequently, the researchers developed an ESCC classification model based on a logit model using HOXC10/HOXD1 methylation status scores, which were validated by Sanger sequencing. The HOXL score effectively identified ESCC from normal samples, with an AUC of 0.96 (95% CI [0.91–0.99]) using an optimal threshold of 0.72, with a sensitivity of 94.8% and specificity of 87.5% (*Yu et al., 2022*). Later, *Qiao et al. (2021)*, analyzed 13 cfDNA samples obtained from the plasma of patients with ESCC using Sanger sequencing for HOXD1 and HOXC10 methylation. The results showed that HOXD1 CpG sites had a methylation rate of 90.77% (118/130), whereas HOXC10 CpG sites had a methylation rate of 60.67% (37/61) (*Yu et al., 2022*). These findings suggest that highly methylated HOXL paralogs, particularly the combination of HOXC10/HOXD1 methylation, have significant potential for the early detection of ESCC.

Researchers have developed early screening models for various cancers, including EC (*Liu et al., 2020a*; *Kandimalla et al., 2021*; *Gao et al., 2023*). *Chen et al. (2020)* from the University of California, USA, described a cancer screening test called PanSeer (based on noninvasive blood tests for circulating ctDNA methylation), which can detect cancer-specific methylation markers in the blood. During the study, the researchers reported preliminary results of PanSeer using plasma samples from 605 asymptomatic individuals, among whom 191 were subsequently diagnosed with gastric, esophageal, or liver cancer. The preliminary results of this study indicated that PanSeer detected five common types of cancer (gastric, esophageal, colorectal, lung, and liver cancers) in 88% (95% CI [80–93%]) of diagnosed patients, with a specificity of 96% (95% CI [93–98%]) and 95% (95% CI [89–98%]) of asymptomatic individuals who were later diagnosed with cancer (*Chen et al., 2020*). The significance of this study lies in PanSeer's ability to identify patients who have already developed cancer but are currently asymptomatic, rather than predicting patients who will develop cancer in the future. However, large-scale longitudinal clinical studies
are necessary to confirm the potential of cfDNA methylation for achieving early cancer detection before routine diagnosis.

*Kandimalla et al. (2021)* conducted a study where they obtained whole-genome 450k DNA methylation data for six gastrointestinal (GI) cancers and adjacent normal tissues from TCGA and GEO (GSE72872). They constructed a GI-targeted bisulfite sequencing panel (gitBS) using differentially methylated regions (DMRs) identified through analysis. This allowed them to retrospectively analyze the cfDNA methylation status of 300 patients with GI cancer and healthy individuals. They constructed and validated three panels for cancer detection using machine-learning algorithms. The panel was optimized to determine the minimum number of DMRs required for optimal detection performance (*Kandimalla et al., 2021*). The strengths of this research include achieving an AUC value of 0.94 for ESCC and 0.90 for EDAC in the GI single cancer detection panel, an AUC value of 0.88 in the GI pan-cancer detection panel, and an accuracy of 0.85–0.95 in the multi-GI cancer trace prediction panel (EpiPanGI Dx) (*Kandimalla et al., 2021*). Additionally, this model requires fewer biomarkers than previous studies (*Klein et al., 2021*), making it cost-effective and suitable for the development of diagnostic panels for large-scale clinical applications. In this study, the first 50 DMRs were sufficient to achieve optimal accuracy in GI single-cancer detection, while the first 150 information-rich DMRs achieved optimal performance in GI pan-cancer and multi-GI cancer classification models (*Kandimalla et al., 2021*).

*Gao et al. (2023)* constructed an early screening model for six cancers, including rectal, esophageal, liver, lung, ovarian, and pancreatic cancers. They built and validated a custom panel of 161,984 CpG sites using public and internal methylation databases. Subsequently, they retrospectively collected cfDNA samples from 1,693 participants to train and validate two different multi-cancer detection blood test models (MCDBT-1/2) under different clinical conditions. Both MCDBT-1 and MCDBT-2 models performed similarly, with MCDBT-1 having a screening specificity of up to 98.9% and a sensitivity of 69.1% in the independent validation cohort. The accuracy of cancer tissue tracing was 82.3%, and the real-world screening sensitivity of the MCDBT-1 model was 70.5%. Widespread adoption of this screening model could reduce the number of late-stage patients with these six common cancers by 38.7–46.4% and increase the relative five-year survival rate by 40%. In other words, implementing comprehensive early cancer screening based on this model could diagnose 38.7–46.4% of patients in the relatively early stages, addressing the current situation where approximately 60% of patients are diagnosed in the later stages. The absolute five-year survival rate of these six cancers could increase from 31.4% to 41.8–44.1% with curative treatments (*Gao et al., 2023*).

Overall, the application of cfDNA methylation in early tumor detection mainly focuses on two aspects: the high methylation of tumor suppressor gene promoter sites and the low methylation of oncogenes. For example, the tumor suppressor genes NRN1, JAM3, and RASSR2 are highly methylated in their promoter regions (*Guo et al., 2016*; *Du et al., 2021*; *Yang et al., 2022*), whereas PAX9, SIM2, and THSD4 are expressed in normal esophageal tissues but are downregulated in tumors (*Talukdar et al., 2021*). Nevertheless, it is crucial to note that the majority of participants in early screening models for EC have already been

diagnosed, which may introduce variability in the sensitivity for undiagnosed cases, posing difficulties in applying these research findings to the real world.

**Based on cfDNA Methylation ongoing trials**

In January 2023, based on the remarkable performance of cfDNA methylation in early cancer diagnosis, the "OverC™ Multi-Cancer Detection Blood Test" developed by Burning Rock Biotech (Irvine, CA, USA) on the ELSA-seq technology platform, was granted the Breakthrough Device Designation by the US FDA, becoming the third globally recognized multi-cancer early detection product with this designation. In May of the same year, Burning Rock Biotech published the latest results from the THUNDER study (NCT04820868) (*Gao et al., 2023*), which revealed the following key findings: (1) The MCDBT-1 model displayed leading international performance in six major types of cancer, including lung, liver, colorectal, ovarian, esophageal, and pancreatic cancers, with a specificity of 98.9% and a sensitivity of 69.1%; (2) The MCDBT-1 model demonstrated superior capabilities in predicting tissue origin, achieving prediction accuracies of 83.2% for TPO1 (primary origin) and 91.7% for TPO2 (primary and secondary origins); (3) In a real-world simulation, the MCDBT-1 model enabled the early diagnosis of 38.7%–46.4% of late-stage cancers (from stages III–IV to I–II), leading to a relative increase in the 5-year survival rate of these six cancer types by 33.1%–40.4%.

Currently, Burning Rock Biotech is collaborating with several clinical research centers on multiple large-scale research cohorts, including (1) PREDICT, the first prospective, multi-cancer early detection clinical trial in China with a cohort size of over ten thousand individuals; (2) PRESCIENT, the first prospective pan-cancer early detection study involving liquid biopsy and multi-omics analysis with a cohort size of ten thousand individuals; and (3) PREVENT, the first prospective, interventional early detection study targeting asymptomatic individuals with a cohort size of ten thousand individuals. The progress of these studies will accelerate the clinical validation of cfDNA methylation for early cancer diagnosis, addressing the need for various cancer screening methods, including esophageal cancer.

## FUTURE PROSPECTS

Detecting cancer signals during the early stages of EC, when symptoms are not yet prominent and the disease has not yet progressed to a late stage, can potentially improve the success of surgical treatment. Recent research has shown significantly elevated cfDNA methylation levels in patients with EC compared to healthy individuals (*Salta et al., 2020*; *Talukdar et al., 2021*), and specific biomarkers unique to EC have been identified (*Li et al., 2019*; *Jammula et al., 2020*). Presently, an early cfDNA methylation screening model has been developed, with a sensitivity of up to 74.7% and a specificity of up to 95.9% (*Qiao et al., 2021*). Machine learning techniques combined with cfDNA methylation sequencing offer a useful approach for early cancer diagnosis. *Zhou et al. (2022)* utilized this approach to decode tumor information and determine the origin of the tumor, achieving a sensitivity of 86.1% and specificity of 94.7% for early cancer detection. Therefore, with the continuous development and improvement of cfDNA methylation sequencing technology, the cfDNA

early screening model we aim to establish can potentially detect asymptomatic cancer using regular blood tests before the diagnosis is confirmed.

While ctDNA methylation analysis has the potential to significantly improve early cancer screening methods and reduce cancer-related mortality, more research is needed to identify the most accurate cfDNA methylation markers suitable for large patient populations and the potential benefits of combining these markers with ctDNA mutation detection. In addition, large-scale clinical studies are necessary to assess the benefits of cfDNA methylation-based early screening models and the impact of early detection on patient survival.

In summary, the development and improvement of cfDNA methylation sequencing technology hold great promise for early detection of cancer in routine clinical settings. These advancements in early screening techniques have the potential to save lives and improve cancer treatment outcomes. Nevertheless, further research is needed to fully understand their impact and potential.

## CONCLUSION

This review focuses on the recent advancements in early detection of esophageal cancer using cfDNA. Although standardized diagnostic methods may not always effectively detect the early stages of the disease, methylation-based detection has shown promise. Non-invasive liquid biopsy approaches significantly simplify the sample collection process, making diagnostic results easier to obtain and generally more reliable. This manuscript includes the latest research designs and their corresponding trial data, commonly using DNA methylation sequencing techniques to demonstrate the utility and effectiveness of cfDNA methylation in the early detection of esophageal cancer. With further clinical trials confirming the advantages of cfDNA methylation in early cancer diagnosis, cfDNA methylation could be incorporated into preventive care, resulting in substantial improvements in early detection of esophageal cancer at a low cost and with increased safety.

### Funding
This study was funded by the Graduate Innovation Project of the Jilin University (No. 2022003), Beijing Science and Innovation Medical Development Foundation (No. KC2022-JX-0025), and Science and Technology Department of Jilin Province (20130604050TC & 20210204123YY). The funders had no role in study design, data collection and analysis, decision to publish, or preparation of the manuscript.

### Grant Disclosures
The following grant information was disclosed by the authors:
The Graduate Innovation Project of the Jilin University: No. 2022003.
Beijing Science and Innovation Medical Development Foundation: KC2022-JX-0025.
Science and Technology Department of Jilin Province: 20130604050TC, 20210204123YY.

## Competing Interests

The authors declare there are no competing interests.

## Author Contributions

- Rui Wang conceived and designed the experiments, analyzed the data, prepared figures and/or tables, and approved the final draft.
- Yue Yang performed the experiments, analyzed the data, authored or reviewed drafts of the article, and approved the final draft.
- Tianyu Lu conceived and designed the experiments, prepared figures and/or tables, and approved the final draft.
- Youbin Cui performed the experiments, prepared figures and/or tables, and approved the final draft.
- Bo Li conceived and designed the experiments, analyzed the data, authored or reviewed drafts of the article, and approved the final draft.
- Xin Liu conceived and designed the experiments, performed the experiments, analyzed the data, prepared figures and/or tables, authored or reviewed drafts of the article, and approved the final draft.

## Data Availability

This is a literature review.

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
