# Peer review of "Circulating cell-free DNA-based methylation pattern in plasma for early diagnosis of esophagus cancer"

_PeerJ, doi:10.7717/peerj.16802_

## Round 0.1 · original submission · Major Revisions

Please in your revision take special note the concerns of reviewer 2, and make special mention of the novelty aspect of this work.

**Language Note:** PeerJ staff have identified that the English language needs to be improved. When you prepare your next revision, please either (i) have a colleague who is proficient in English and familiar with the subject matter review your manuscript, or (ii) contact a professional editing service to review your manuscript. PeerJ can provide language editing services - you can contact us at [email protected] for pricing (be sure to provide your manuscript number and title). – PeerJ Staff

Reviewer 1 ·

Basic reporting

No comment

Experimental design

For the study method - please add the date range of articles considered. Would also be helpful to include the total number of articles reviewed (the ones related to your specific aims of review).

Most of the sources are appropriately cited and paraphrased with the following exception:
Line 205 - "Studies have demonstrated(Guo et al. 2017) MBs with a size of 95 bp and comprising more than three CpG sites are more readily detectable in circulating cfDNA of approximately 170 bp." Is this supposed to read "CpG sites are more readily detectable than circulating cfDNA of approximately 170 bp" - In reviewing the source I am not clear where the 170 bp comparison came from

Line 211-214: This paraphrasing is very similar in wording to the abstract of the cited source. May change the phrases a bit. In the following sentence "The study found that LINE-1 sequences in EADC cfDNA exhibited low levels of methylation, and the longitudinal analysis revealed a correlation between the methylation status of LINE-1 sequences in cfDNA and the progression to EADC (Boldrin et al. 2020)" There were low levels of methylation in the Barretts samples and these increased with progression to EADC (currently reads that EADC progresses to EADC).

overall structure and organization is excellent

Validity of the findings

No comment

Additional comments

Line 46 - "most EC patients cannot cure cancer by surgical resection" - Confusing wording - would change this to "most EC patients cannot be cured with surgical resection."
line 65 - CCGA - please define abbreviation
Line 105 - "Therefore, computer learners and patients with EC may also be potential readers for this study" _ this is not written as a patient resource so recommend removing if authors agree
Line 156 - Please define EMSO

Reviewer 2 ·

Basic reporting

The manuscript is consisting of summaries of various existing studies without adding any new insights. For a manuscript to be valuable and contribute to the field, it should ideally offer a unique perspective, synthesis of existing research, and potentially propose novel ideas or conclusions based on the existing studies.

Experimental design

no comment.

Validity of the findings

no comment.

Reviewer 3 ·

Basic reporting

General Comments
The topic of circulating cell-free DNA (cfDNA) methylation in early diagnosis of esophageal cancer is timely and important. Early detection of cancer is crucial for effective treatment, and the abstract suggests that cfDNA methylation could be a promising biomarker for this purpose.

Major Points
Scope and Focus: While the abstract outlines the general areas the manuscript will cover, it would be beneficial to understand the exact scope of the review. Are you covering all methods for analyzing cfDNA methylation, or is the focus primarily on next-generation sequencing techniques, for example?

Clarity and Organization: Consider reorganizing the sections of the paper to enhance the logical flow of content. Given your own comments on section headers, the inclusion of well-defined sections like "Introduction," "Methods in cfDNA Methylation Analysis," "Clinical Applicability," and "Future Prospects" could be beneficial.


Clinical Trials: The need for large-scale prospective clinical trials is mentioned, which is great. However, are there any ongoing trials that readers should be aware of? Any preliminary results from such trials?

Technological Advancements: You mention "recent advancements in separation and detection techniques" for cfDNA methylation analysis. Could you elaborate on what these advancements are?


Conclusion
This manuscript holds promise to serve as a comprehensive review on the role of cfDNA methylation patterns in the early detection of esophageal cancer. However, it requires some reorganization and elaboration on key aspects to make it a valuable resource in the field.

I look forward to seeing the revisions.

Experimental design

No comments.

Validity of the findings

No comments.

Additional comments

No comments.

---

## Round 0.2 · accepted · Accept

Thanks for the revised manuscript, the reviewers indicated that all concerns were addressed.

Reviewer 1 ·

Basic reporting

no comment

Experimental design

reviewers have addressed all items brought up from the initial review

Validity of the findings

no comment

Additional comments

Reviewers have appropriately responded to comments from review

Reviewer 3 ·

Basic reporting

With the revised manuscript, authors have addressed all my concerns.

Experimental design

No comments.

Validity of the findings

No comments.

Additional comments

No comments.